# NATIVQA: MULTILINGUAL CULTURALLY-ALIGNED NATURAL QUERIES FOR LLMS

## ABSTRACT

Natural Question Answering (QA) datasets play a crucial role in evaluating the capabilities of large language models (LLMs), ensuring their effectiveness in real-world applications. Despite the numerous QA datasets that have been developed, there is a notable lack of region-specific datasets generated by native users in their own languages. This gap hinders the effective benchmarking of LLMs for regional and cultural specificities. Furthermore, it also limits the development of fine-tuned models. In this study, we propose a scalable, language-independent framework, *NativQA*, to seamlessly construct culturally and regionally aligned QA datasets in native languages, for LLM evaluation and tuning. We demonstrate the efficacy of the proposed framework by designing a multilingual natural QA dataset, Multi*NativQA*, consisting of ∼64k manually annotated QA pairs in seven languages, ranging from high to extremely low resource, based on queries from native speakers from 9 regions covering 18 topics. We benchmark open- and closed-source LLMs with the Multi*NativQA* dataset. We also showcase the framework efficacy in constructing fine-tuning data especially for low-resource and dialectally-rich languages. We made both the framework *NativQA* and Multi*NativQA* dataset publicly available for the community.[1]

## 1 INTRODUCTION

Recent advancements in LLMs have revolutionized the landscape of artificial intelligence, significantly pushing the state-of-the-art for a broad array of Natural Language Processing (NLP) and Speech Processing tasks, such as machine translation (MT), Question Answering (QA), automatic speech recognition, among others. Their potential in language understanding and generation, across multiple (high- and low-resourced) languages, has attracted researchers to integrate and benchmark the LLM capabilities across diverse tasks, domains, and disciplines (OpenAI, 2023; Touvron et al., 2023). However, the rapid integration of LLMs necessitates measuring cultural discrepancies in the responses generated by LLMs to ensure alignment with users' cultural values and contexts (Naous et al., 2024; AlKhamissi et al., 2024; Shen et al., 2024; Liu et al., 2024; Arora et al., 2024; Myung et al., 2024). This is particularly crucial in cross-lingual scenarios, where LLMs hallucinate or produce stereotypical responses biased toward Western culture, neglecting diverse cultural norms (Naous et al., 2024). Consequently, such biases hinder the effectiveness of LLMs in daily-use applications for diverse languages and cultures, largely due to their under-representation in the training data used for these models.

There is a notable absence of such multilingual region-specific cultural benchmarks designed to evaluate the LLMs' performance across different cultures and languages. As a result, multilingual and non-English LLMs have been evaluated by using MT, with or without human involvement, to translate the existing English datasets into corresponding languages. For instance, to assess models like Jais (Sengupta et al.,

---

[1] https://nativqa.gitlab.io/

| Lang | Q/A | Example (Native) | English Translation |
|------|-----|------------------|---------------------|
| Arabic | Q | كم مساحة قطر طول وعرض؟ | What is the area of Qatar length and width? |
| | A | يبلغ عرض مساحتها حوالي 100 كم وتمتد بطول 200 كم في الخليج. | Its area is about 100 km in width and extends 200 km in the Gulf. |
| Assamese | Q | কোন জন বিখ্যাত ৰাজনৈতিক ব্যক্তিয়ে শেহতীয়াকৈ অসমত বিজেপিৰ পৰা কংগ্ৰেছলৈ যোগদান কৰিছিল ? | Which famous political person recently joined from BJP to Congress in Assam? |
| | A | আমিনুল হক লস্কৰে শেহতীয়াকৈ অসমত বিজেপিৰ পৰা কংগ্ৰেছত যোগদান কৰিছিল। | Aminul Haque Laskar recently joined Congress from BJP in Assam. |
| Bangla | Q | শোলাকিয়া মাঠের আয়তন কত ? | What is the area of Sholakia field? |
| | A | বর্তমান শোলাকিয়া ঈদগাহ মাঠের আয়তন ৭ একর। | The current area of Sholakia Eidgah field is 7 acres. |
| English | Q | Does UDST offer scholarships? | NA |
| | A | Public schools in Qatar receive government funding and provide free tuition to all citizens. | NA |
| Hindi | Q | नवरात्रि में कलश रखने का शुभ मुहूर्त क्या है? | What is the auspicious time to keep Kalash in Navratri? |
| | A | कलश की स्थापना चैत्र शुक्ल पक्ष की प्रतिपदा तिथि को की जाती है. इस बार चैत्र नवरात्रि की घटस्थापना का सबसे अच्छा मुहूर्त सुबह 6 बजकर 2 मिनट लेकर सुबह 10 बजकर 15 मिनट तक है | The Kalash is established on the Pratipada date of Chaitra Shukla Paksha. This time the best time for Chaitra Navratri is from 6.02 am to 10.15 am. |
| Nepali | Q | नेपालको सबैभन्दा ठूलो ताल कुन हो | Which is the biggest lake in Nepal? |
| | A | नेपालको सबैभन्दा ठूलो ताल कर्णाली प्रदेशको रारा ताल हो। | The largest lake in Nepal is Rara Lake in Karnali Province. |
| Turkish | Q | İstanbul'da göl var mı? | Is there any lake in Istanbul? |
| | A | İstanbul'da dört doğal göl bulunmaktadır. Bunların yanı sıra, baraj gölleri de vardır. | There are four natural lakes in Istanbul. In addition, there are also reservoir lakes. |

Figure 1: Examples of questions and answers in different languages with their translation from our dataset.

2023) and AceGPT (Huang et al., 2024), evaluation datasets have been translated into Arabic. Other efforts include Korean MMLU (Son et al., 2024) and Okapi (Lai et al., 2023b) where the authors translated three benchmark datasets in 26 languages. However, adopting the translation process often fails to capture the rich regional and cultural nuances embedded within the target languages. The typical alternative of translation is to develop datasets in new languages by human annotators, which is a costly and time-consuming process. In a recent study, Arora et al. (2024) developed 1.5K culture-specific QAs by gathering questions from community web forums and employing native speakers to manually write questions. Similarly, Myung et al. (2024) produced 52.5K multiple-choice and short-answer questions, with both question collection and answer writing being fully manual.

In this study, we propose a framework, **Nativ**e QA (*NativQA*), specifically designed to seamlessly develop regionally- and culturally- specific QA datasets following a human-machine collaborative approach. Datasets developed through *NativQA* serve two primary functions: *(i)* evaluating the LLM performance over real users' information needs and interests expressed in their native languages, and *(ii)* facilitating fine-tuning of LLMs to adapt to cultural contexts. Moreover, to show the efficacy of the *NativQA* framework, we developed a natural **Multi**lingual **Nativ**e question-answering (**QA**) dataset, Multi*NativQA*, including $\sim 64k$ QA pairs in 7 extremely low to high resource languages (see in Figure 4), covering 18 different topics from 9 different regions (see examples in Figure 1).

We further demonstrate the usefulness of both *NativQA* framework and Multi*NativQA* dataset by fine-tuning open LLMs. Fully fine-tuning LLMs is computationally expensive due to large number (billions or even trillions) of learnable parameters (Fedus et al., 2022). Hence, we adopted parameter-efficient fine-tuning (PEFT) (Liu et al., 2023; Houlsby et al., 2019; Hu et al., 2022), which only update a small number of parameters, significantly reducing the computational cost.

Unlike Arora et al. (2024); Myung et al. (2024), the proposed *NativQA* framework can seamlessly collect QA pairs with minimal human intervention. Additionally, the answers are grounded in web-based reference sources. Our approach is inspired by the regional-based search engine queries addressing everyday needs as shown in Figure 3. Therefore, our contribution in this study are as follows:

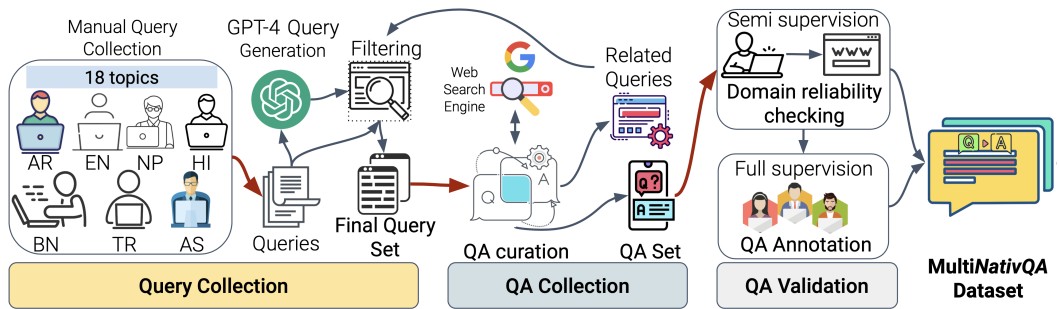

Figure 2: *NativQA* framework, demonstrating the data collection and annotation process.

- We propose the semi-automatic – *NativQA* framework for developing culture- and region-specific natural QA datasets, enhancing LLMs inclusivity and providing comprehensive, culturally aligned benchmarks.
- We develop and release the Multi*NativQA* dataset, in seven languages with $\sim 64k$ manually annotated QA pairs, covering 18 different topics from native speakers across 9 different regions. Additionally, we release another $55k$ QA pairs from six different locations developed using our semi-supervised approach.
- We benchmark over Multi*NativQA* with 2 open and 2 closed LLMs, advancing research in this area.
- We report experimental results of a fine-tuned Llama-3.1 model across all languages.

Our findings emphasize the importance of well-crafted benchmarks efforts for studying regional/cultural awareness in LLMs. The results supports the hypothesis that under-represented regions, and dialectal-rich language (e.g., Arabic) benefit more from incorporating native and culturally aware information in the LLM. This highlights the value of proposed language-independent framework *NativQA*, which efficiently create such multilingual, region and cultural-specific resources with minimal human effort.

## 2 RELATED WORK

LLMs have consistently showcased impressive capabilities spanning diverse disciplines and tasks. Hence, there have been efforts to evaluate the performance of LLMs on standard NLP tasks (Bubeck et al., 2023; Bang et al., 2023; Ahuja et al., 2023; Hendy et al., 2023). While there have been several efforts to develop resources and benchmark LLMs with those resources, most of the prior works are limited to English. Furthermore, regarding the evaluation for other languages, translated forms are commonly used Lai et al. (2023b); Sengupta et al. (2023); Huang et al. (2024).

**Existing QA Datasets** Question Answering has been a standard NLP task for decades, pushing the development of many QA datasets in different languages. Kwiatkowski et al. (2019) and Yang et al. (2018) proposed two extractive QA datasets including Natural Questions (NQ), both containing long-form answers for questions that include large-scale question-answer pairs. The generated long answer's faithfulness is estimated by measuring the ratio of the golden short answer span contained in it. Joshi et al. (2017) developed TriviaQA dataset, which consists of 650k question-answer-evidence triples. These triples are created by merging 95K question-answer pairs. Rajpurkar et al. (2016) developed SquAD, which is a collection of 100k crowdsourced questions and answers paired with shortened Wikipedia articles. HelpSteer (Wang et al., 2023) is another QA dataset, which comprises a 37k sample dataset with multiple attributes of helpfulness preference that covers verbosity, accuracy, coherence, and complexity in addition to overall helpfulness. The most closest work in the literature to ours is BLEND Myung et al. (2024) which is a hand-crafted benchmark consisting of 52.6k multiple choice and short-answer QA pairs for 13 different languages in total, focusing cultural aspects of languages.

**Evaluations of LLMs for QA** For LLM evaluation, there are notable datasets covering world knowledge (Hendrycks et al., 2020), commonsense reasoning (Zellers et al., 2019), reading comprehension (Bandarkar et al., 2024), factuality (Lin et al., 2022), and others. These datasets are usually transformed into multiple-choice questions. Additionally, standard QA datasets have also been used for LLM evaluation (Hu et al., 2020). Kamalloo et al. (2023) performed the analysis of different open-domain QA models, including LLMs by manually judging answers on a benchmark dataset of NQ-open (Lee et al., 2019), and reported a systematic study of lexical matching. Their investigation shows that LLMs attain state-of-the-art performance but fail in lexical matching when candidate answers become longer. In Table 4 (Appendix), we report the most notable existing QA datasets compared to ours. Compared to existing datasets, Multi*NativQA* dataset is novel in terms of its topical coverage with a focus on cultural aspects, and being regionally-native.

## 3 NativQA Framework

Figure 2 presents the *NativQA* framework with three inter-connected modules described below.

### 3.1 Query Collection (QC)

The objective of this module is to collect open-ended queries, $\varrho$, centered on various predetermined topics derived from common concepts in everyday communication. The topic set is first manually constructed. This manual effort allows us to identify topics that are culture- or region-specific. Examples of seed topics include: *Animals, Business, Clothing, Education, Events, Food & Drinks, General, Geography, Immigration, Language, Literature, Names & Persons, Plants, Religion, Sports & Games, Tradition, Travel*, and *Weather*.

Following, we start collecting the manual query set $\varrho_m$. We began by recruiting native speakers of the language of the target countries. Each speaker is encouraged to write $m$ queries per topic, in their native or second language,[2] focusing on queries they might ask a search engine as residents of a corresponding major city. We then expand the $\varrho_m$ set with synthesized queries, $\varrho_s$. Synthesizing queries helps to increase the variability in sub-topics and improve the versatility of writing styles in the final set of queries. For $\varrho_s$, we prompted an LLM to generate $x$ similar queries for each input query, $\varrho_m^i \in \varrho_m$. Finally, $\varrho_s$ is de-duplicated against $\varrho_m$ using exact string matching, resulting in the *final set* of seed queries, $\varrho_0 = \varrho_m \bigcup \varrho_s$.

### 3.2 QA Collection (QAC)

Next, leveraging a search engine, we automatically collect QA pairs that potentially cover queries $\varrho_0$. We specifically selected 'Google', due to its feature – "People also ask", where it lists several questions, searched by real users and are potentially relevant to the initial user query, as shown in Figure 3. Moreover, these questions $Q$ are associated with answers $A$ extracted by the search engine, along with the attribution, $L$ – links to the sources of the answers.

Our QA curation module implements Algorithm 1, using the seed queries $\varrho_0$ along with the number of iteration, $N_{iter}$, as input. For each iteration $i \in N_{iter}$, we collect QA pairs $P_{QA}^i$, and related queries $S\varrho_{rel}^i$ for each query, $q \in S\varrho$, and then pass it to the filtering module and update the current query set $S\varrho$. We repeat the process for all iterations to obtain the final QA set, $S_{QA}$ with enriched queries $S\varrho$.

### 3.3 QA Validation (QAV)

Following, we validate the extracted QA pairs, considering at least two aspects: *(i)* the quality and answerability of questions, and *(ii)* reliability and completeness of answers. We validate the QA pairs through the following steps.

---

[2]widely used in the respective city

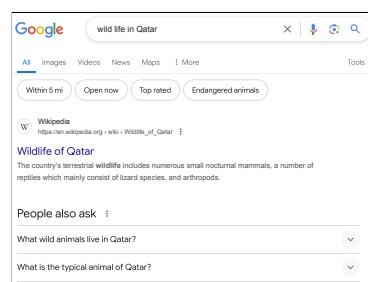

Figure 3: Google's QA list in response to a query.

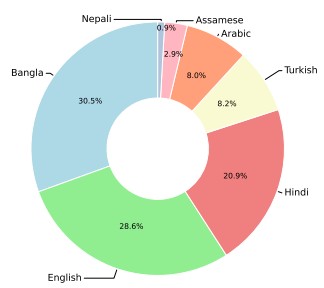

Figure 4: Distribution of the Multi*NativQA* dataset across different languages.

**Algorithm 1** Collecting QA pairs using seed queries $\varrho_0$. $P_{QA}^i$: QA pair, $S\varrho_{rel}^i$: related queries. ExtractQA(*) and ExtractRelatedQueries (*) are functions that return questions, $Q$- answers, $A$ pairs with attribution $L$, and related queries, respectively, which are obtained from the search engine for a given query, $q$. DeDuplication (*) removes any duplicate entries from the set to ensure uniqueness.

1: **Input:**
2:     Seed queries: $\varrho_0 = \{\hat{\varrho_1}, \hat{\varrho_2}, \ldots, \hat{\varrho_m}\}$
3:     Number of iterations: $N_{iter}$
4: **Output:**
5:     Set of QA pairs: $S_{QA}$
6:     Set of enriched queries: $S\varrho$
7: $S_{QA} \leftarrow \emptyset$
8: $S\varrho \leftarrow \varrho_0$
9: **for** $i$ from 1 to $N_{iter}$ **do**
10:     $P_{QA}^i \leftarrow \emptyset$
11:     $S\varrho_{rel}^i \leftarrow \emptyset$
12:     **for** $q \in S\varrho$ **do**
13:         $(Q^q, A^q, L^q) \leftarrow$ ExtractQA($q$)
14:         $P_{QA}^i \leftarrow P_{QA}^i \cup \{(q', a', l') \mid q' \in Q^q, a' \in A^q, l' \in L^q\}$
15:         $S\varrho_{rel}^i \leftarrow S\varrho_{rel}^i \cup$ ExtractRelatedQueries($q$)
16:     **end for**
17:     $P_{QA}^i \leftarrow$ DeDuplication($P_{QA}^i$)
18:     $S_{QA} \leftarrow S_{QA} \cup P_{QA}^i$
19:     $S\varrho \leftarrow S\varrho \cup S\varrho_{rel}^i$
20: **end for**
21: **return** $S_{QA}, S\varrho$

**Domain Reliability Check (DRC).** First, we extracted a unique set of web-domains using the attribution[3], $L$ from the extracted QA pairs, $S_{QA}$. We then manually classify each domain's reliability based on an annotation guideline specifically designed for this task, inspired by several relevant studies (Selejan et al., 2016; Flanagin & Metzger, 2007; Metzger & Flanagin, 2015). Next, we filtered out the QA pairs to retain answers only from annotated reliable sources as we hypothesize that answers from web pages on reliable domains are likely to be trustworthy. We adapted this approach as it offers more practical and scalable solution by reducing manual effort required to obtain more reliable QA pairs. The final compiled list of reliable domains (e.g., BBC, Guardian) can be further utilized to extract QAs for new queries for multiple languages, specially when developing fine-tuning data.

**QA Annotation (QAA).** Although some domains are considered reliable, the content they host may not always be trustworthy due to unreliable user-generated content. To address this, we further refined our framework by manually checking and editing the curated QA pairs from reliable sources. For each QA pair, we apply four types of annotations. *(i) Question validation*: Human annotators verify questions' quality by classifying each question as "Good question" or "Bad question". We then proceed to the subsequent steps using only the questions classified as "Good" . *(ii) Question's relevancy to the location*: Annotators are asked to classify whether the question is related to the specified location. *(iii) Answer categorization*: Annotators

---
[3]answer-source links

examine each QA pair and assess whether the answer provides sufficient information to satisfy the question, and categorize the answers based on the correctness (see Section 4.2.2). *(iii) Answer editing*: If an answer does not address all parts of a question, or wrong, annotators must edit the answer using content from the answer's source Web page. We limit the annotators to using the provided source Web pages to maintain the scope and the reliability of answers we collect during this phase. A detailed annotation guideline for the above steps is provided in the Appendix D.3.

## 4   MULTI*NativQA* DATASET

We demonstrate the effectiveness and scalability of the *NativQA* framework by creating a large-scale, multi-lingual Multi*NativQA* dataset. The Multi*NativQA* dataset spans over 7 languages – from high- to extremely low-resource and 9 different location/cities. Multi*NativQA* captures linguistic diversity, by including several dialects for dialect-rich languages like Arabic.[4] We also added two linguistic variations of Bangla to reflect differences between speakers in Bangladesh and West Bengal, India. Furthermore, we included English queries from Dhaka and Doha, where English is often used as a second language.

### 4.1   *NativQA* FRAMEWORK ADAPTATION

**Query Collection** For multilingual QC, we started with predetermined topics (see Section 3.1) derived from common concepts in everyday lives of users (details in Appendix D.1). Next, we asked the residents and the native speakers to write 10 to 50 queries[5] per topic about their major cities and urban areas. We then used GPT-4 to generate 10 similar queries based on each input query and applied de-duplication on the seed queries. The resultant number of queries per region are reported in Table 1.

**QA Collection** Using *QAC Module* we enriched queries and QA pairs for each language and its respective city. We ran our collection algorithm for 3-7 $N_{iter}$ per region based on the convergence rate. We collected $\sim 154K$ QA pairs across all languages (see Table 1:#QA).

**QA Validation** The *QAV* is the final (and optional) phase of the *NativQA* framework. It includes two steps: domain reliability check (DRC) and QA annotation (QAA). These steps ensures high quality of the dataset and can be executed to the entire dataset or only test split, depending on the cost and time constraints.For Multi*NativQA*, we executed both the DRC and QAA steps to all target languages and regions to create a high-quality resource for the research community (see Section 4.2).

### 4.2   MANUAL ANNOTATION

We briefly discuss the manual annotation effort for QAV phase in *NativQA* framework for developing Multi*NativQA* dataset. For more detail instruction and analysis see Appendix D.2.

#### 4.2.1   DOMAIN RELIABILITY CHECK

The objective for the domain reliability check is to verify the credibility of the source domain, which can be used to judge the factuality and reliability of answers sourced from that domain. We adopt the following definition of the credibility of the domain/website: "A credible webpage is one whose information one can accept as the truth without needing to look elsewhere. If one can accept information on a page as true at face value, then the page is credible; if one needs to go elsewhere to check the validity of the information on the page, then it is less credible" (Schwarz & Morris, 2011). Annotators were tasked to review each web

---

[4]Besides the formal Modern Standard Arabic (MSA), we added six Arabic dialects—Egyptian, Jordanian, Khaliji, Sudanese, Tunisian, and Yemeni – to capture Doha's linguistic and cultural diversity.

[5]Without a strict limit, some topics exceeded 50 queries.

Table 1: Statistics of our Multi*NativQA* dataset including languages with initial seed queries, the number of QA pairs collected per language from different locations and the final annotated QA pairs. CC: Country code, Lang.: Language, SQ: Seed Query, Cat.: Categorization in terms of high (H), medium (M), low (L), and extremely low (X) as per Lai et al. (2023a), – Only testing split due to limited dataset size.

| Lang. | Cat. | City | C.Code | # of SQ | # of QA | # Final Annotated QA | | | |
| | | | | | | Train | Dev | Test | Total |
|---|---|---|---|---|---|---|---|---|---|
| Arabic | M | Doha | QA | 3,664 | 12,311 | 3,649 | 492 | 988 | 5,129 |
| Assamese | X | Assam | IN | 900 | 21,009 | 1,131 | 157 | 545 | 1,833 |
| Bangla | L | Dhaka | BD | 889 | 13,688 | 7,018 | 953 | 1,521 | 9,492 |
| Bangla | L | Kolkata | IN | 900 | 13,378 | 6,891 | 930 | 2,146 | 9,967 |
| English | H | Dhaka | BD | 1,339 | 17,744 | 4,761 | 656 | 1,113 | 6,530 |
| English | H | Doha | QA | 3,414 | 25,621 | 8,212 | 1,164 | 2,322 | 11,698 |
| Hindi | M | Delhi | IN | 1,184 | 16,328 | 9,288 | 1,286 | 2,745 | 13,319 |
| Nepali | L | Kathmandu | NP | 1,222 | 11,503 | – | – | 561 | 561 |
| Turkish | M | Istanbul | TR | 900 | 23,143 | 3,527 | 483 | 1,218 | 5,228 |
| **Total** | | | | **14,412** | **154,725** | **44,477** | **6,121** | **13,159** | **63,757** |

domain to determine its credibility and assign one of the following four reliability labels: *(i)* very reliable, *(ii)* partially reliable, *(iii)* not sure, *(iv)* completely unreliable.

### 4.2.2 QA ANNOTATION

This step of the QAV involves four types of annotations. Below, we discuss the brief guidelines for each annotation.

1. **Question validation:** The purpose of this task is to evaluate the quality of the questions. The annotators classified whether the questions are "Good" or "Bad" based on the criteria discussed below. The choice of the two types of questions was inspired by the NQ dataset (Kwiatkowski et al., 2019). Depending on the annotation, the annotator's subsequent tasks vary. If a question is marked as 'good', they proceed to the next task for the QA pair; otherwise, they skip further annotation and move on to the next QA pair.
2. **Question's relevancy to the location:** The purpose of this annotation was to check whether the question is related to the location it was intended to collect. For example, "*What is the main city in Qatar?*" is a question related to Qatar.
3. **Answer categorization:** An answer can be categorized into one of these categories: *(i)* correct, *(ii)* partially correct, *(iii)* incorrect, and *(iv)* the answer can't be found in the source page. Complete definition for each category is provided in Appendix D.3.
4. **Answer editing:** This step ensures the answer is correct, fully responds to the question, and is fluent and informative. If the answer is incorrect or incomplete, annotators must check the source page to extract content that completes the answer, if available.

### 4.3 ANNOTATION TASK SETUP

The annotation team consisted of native speakers of the respective languages, with English as their second language. The annotators had diverse educational backgrounds, ranging from undergraduate students to those holding PhD degrees. The team was trained and monitored by language specific expert annotators. To ensure quality, periodic checks of random annotation samples were conducted, and feedback was provided. Three annotators were assigned to the DRC task, and the final label is assigned based on majority voting. For the QAA task, each QA pair was annotated by two annotators for the test set. In cases of disagreement, a third annotator reviewed and revised the annotations. For the training and dev set, each QA pair was annotated by one annotator. These choices were made to maintain a balance between annotation quality, time, and cost. We utilized in-house annotation platform for the tasks as discussed in Appendix D.6.

Table 2: Performance of different LLMs across languages. F1: F1 BERTScore, Rou.: Rouge1, Llama-3.1: Llama-3.1-8B-Instruct, Gemini-1.5: Gemini-1.5 Flash, Mistral: Mistral- 7B-Instruct-v0.1. **Bold** results are best per column per language. *Italicized* results are best across open models. **AVG** Average over languages.

| Model | F1 | BLEU | Rou. | F1 | BLEU | Rou. | F1 | BLEU | Rou. | F1 | BLEU | Rou. | F1 | BLEU | Rou. |
|---|---|---|---|---|---|---|---|---|---|---|---|---|---|---|---|
| | **Arabic** | | | **Bangla-IN** | | | **English-BD** | | | **Hindi** | | | **Turkish** | | |
| GPT-4o | 0.839 | **0.280** | **0.044** | 0.821 | 0.226 | 0.009 | **0.651** | **0.384** | **0.284** | **0.865** | **0.296** | **0.050** | 0.768 | **0.226** | **0.252** |
| Gemini-1.5 | **0.840** | 0.228 | 0.038 | **0.833** | **0.251** | **0.014** | 0.631 | 0.259 | 0.251 | 0.800 | 0.171 | 0.036 | **0.773** | 0.164 | 0.229 |
| Llama-3.1 | *0.528* | *0.202* | *0.037* | *0.453* | *0.132* | *0.007* | *0.636* | 0.280 | *0.256* | *0.604* | *0.260* | *0.035* | *0.616* | *0.217* | *0.202* |
| Mistral | 0.487 | 0.148 | 0.034 | 0.418 | 0.108 | 0.005 | 0.620 | *0.345* | 0.251 | 0.553 | 0.177 | 0.030 | 0.563 | 0.193 | 0.161 |
| | **Assamese** | | | **Bangla-BD** | | | **English-QA** | | | **Nepali** | | | **AVG** | | |
| GPT-4o | 0.745 | 0.107 | **0.021** | 0.826 | 0.154 | 0.007 | **0.628** | 0.314 | **0.260** | **0.873** | 0.086 | 0.003 | 0.779 | **0.230** | **0.103** |
| Gemini-1.5 | **0.808** | **0.150** | 0.016 | **0.844** | **0.292** | **0.010** | 0.620 | 0.274 | 0.241 | **0.873** | **0.244** | **0.005** | **0.780** | 0.226 | 0.093 |
| Llama-3.1 | *0.523* | *0.029* | *0.005* | *0.840* | *0.119* | *0.005* | *0.622* | 0.294 | *0.247* | *0.582* | *0.138* | *0.002* | *0.600* | *0.186* | *0.088* |
| Mistral | 0.485 | 0.020 | 0.003 | 0.820 | 0.080 | 0.005 | 0.608 | *0.332* | 0.236 | 0.504 | 0.056 | 0.002 | 0.562 | 0.162 | 0.081 |

## 4.4 ANNOTATION AGREEMENT

We evaluate the Inter-Annotator Agreement (IAA) of manual annotations using the Fleiss' Kappa coefficient ($\kappa$) for the domain reliability tasks. The Kappa ($\kappa$) values across the languages ranges from 0.52 to 0.66 (except for English being **0.37**) which correspond to fair to substantial agreement (Landis & Koch, 1977). Note that we selected the final label where the majority agreed, meaning that we have above 66% agreement on the final label. For the *QA annotation task* (answer editing), we first directly select only the questions where both annotators agree. For the disagreed cases, another annotator revises them; ultimately, we select based on the agreement of at least two annotators. For the answer editing, on average this matching is 66.04% across languages. In addition we have computed Levenshtein distance to understand how much edits has been done. The average edits across all languages are relatively low (0.17), which indicates minimal edits has been done on the answers. In Section H, we provide further details.

## 4.5 STATISTICS AND ANALYSIS

Figure 4 reports the initial data distribution across languages, irrespective of the country they were collected from. English, Arabic, and Bangla are higher in proportion due to the fact that *(i)* English consists of data collected from Qatar and Bangladesh, *(ii)* Arabic consists of queries from different dialects, and *(iii)* Bangla consists of data from Bangladesh and India. The average length for question and answer are 6 and 35 words, respectively. As Table 1 shows, our annotation process resulted in a decrease in QA set size by half (comparing initial QA set (column *#QA*) to final QA set (column *F.QA*)). We also faced a significant drop for Assamese and Nepali. This drop is due to the fact that the search engine returned QA pairs in non-native languages (in these cases, either Hindi or English) rather than the native language. As part of our process, we filtered out QA pairs that are not in the target language. We identify the native language using a language detection tool[6] and then manually revise them. Our final Multi*NativQA* dataset covered a wide range of topics in all languages with similar distribution (see Appendix Figure 6 and Figure 7). To assess the efficacy of the *NativQA* framework, we additionally collected $55k$ QA pairs from 6 different locations, which will be released without any labeling, for the community (see in Appendix F).

## 5 EXPERIMENTAL SETUP

**Data Splits** We split the data for each region into training (70%), development (10%), and test (20%) sets using stratified sampling based on topics as labels. Given the small size of the Nepali data, we kept the

---

[6]http://fasttext.cc/docs/en/language-identification.html

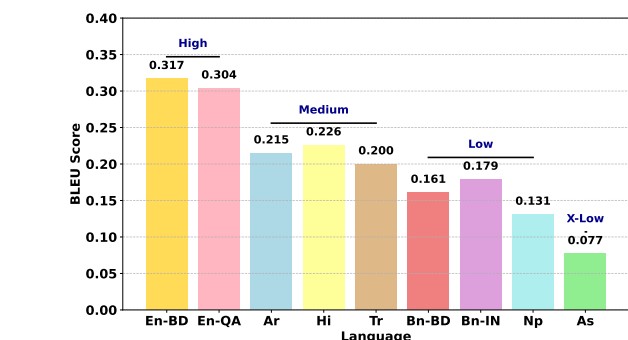

Figure 5: Average performance (BLEU scores) of the models by language. X-Low: Extremely low.

full dataset for test purpose. Note that annotation was done on each data split separately, and some data was discarded because questions were labeled as bad or answers were labeled as incorrect. As a result, the original proportion of the splits is not consistent across languages and data splits (see Table 1).

**Models** We experiment with both open and close LLMs. For the close models we use GPT-4o (Achiam et al., 2023) and Gemini 1.5 Flash.[7] For open models, we opt for Llama-3.1-8B-Instruct,[8] and Mistral-7B-Instruct-v0.1.[9] We use zero-shot learning as our setup with all models. For reproducibility, we set the temperature to zero, and designed the prompts using concise instructions, as reported in Appendix E.1.

**Fine-tuning Models.** We demonstrate the efficacy of Multi*NativQA* training split for all regions by fine-tuning an open LLM – Llama-3.1-8B-Instruct model. To reduce the computational cost, we opt for PEFT using LoRA (Hu et al., 2022). We train the model in full precision (FP16). We use Adam optimizer, set the learning rate to $2e-4$, lora alpha to 16, lora $r$ to 64, maximum sequence length to 512, with a batch size of 16. We fine-tuned the model for one epoch with no hyper-parameter tuning.

**Fine-tuning Instructions.** For fine-tuning, we created a diverse set of English instructions using template-based approach. We design the templates by prompting two close models: GPT-4o and Claude-3.5 Sonnet,[10] to generate 10 diverse instructions per model for the QA task for each language. Following, during fine-tuning, we randomly select one from these templates and append to the QA pair to create the final instruction. During inference, we randomly select one instruction and use it to prompt both the base and the fine-tuned model. Examples of instructions and prompts are in Appendix E.2.

**Evaluation and Metrics.** We utilized the LLMeBench framework (Dalvi et al., 2024) to evaluate the LLMs with Multi*NativQA* test set. For performance measure, we used standard metrics commonly used for QA evaluation. We selected lexical (n-gram) similarity based metrics – BLEU, and ROUGE; and semantic similarity metric – F1 within the BERTScore (Zhang et al., 2020), computed using the contextual embeddings extracted from pre-trained BERT model. We extracted embeddings from language specific transformer models (see Appendix, Table 21).

# 6 RESULTS

**Open *vs* Close LLMs** We reported the performance of both open- and closed-LLMs across all the regions in Table 2. Our result indicates that the closed models (e.g., GPT-4o BLEU-AVG:0.230), outperform the

---

[7]gemini-1.5-flash-preview-0514

[8]https://huggingface.co/meta-llama/Llama-3.1-8B-Instruct

[9]https://huggingface.co/mistralai/Mistral-7B-Instruct-v0.1

[10]https://www.anthropic.com/news/claude-3-5-sonnet

Table 3: Performance of fine-tuned Llama-3.1 model for different languages. Llama-3.1: Llama-3.1-8B-Instruct, Llama-3.1-FT: Fine-tuned.

| Model | F1 | BLEU | Rou. | F1 | BLEU | Rou. | F1 | BLEU | Rou. | F1 | BLEU | Rou. | F1 | BLEU | Rou. |
|---|---|---|---|---|---|---|---|---|---|---|---|---|---|---|---|
| | **Arabic** | | | **Bangla-IN** | | | **English-BD** | | | **Hindi** | | | **Turkish** | | |
| Llama-3.1 | 0.508 | 0.080 | 0.032 | **0.451** | 0.054 | 0.005 | **0.621** | **0.247** | **0.234** | **0.606** | 0.123 | **0.038** | **0.613** | 0.092 | 0.188 |
| Llama-3.1-FT | **0.532** | **0.181** | **0.039** | 0.421 | **0.139** | **0.012** | 0.612 | 0.198 | 0.205 | 0.521 | **0.159** | 0.024 | 0.592 | **0.189** | **0.190** |
| | **Assamese** | | | **Bangla-BD** | | | **English-QA** | | | **Nepali** | | | **AVG** | | |
| Llama-3.1 | 0.550 | 0.020 | 0.006 | **0.841** | 0.037 | 0.004 | **0.603** | **0.202** | **0.218** | **0.591** | 0.103 | 0.002 | **0.598** | 0.107 | **0.081** |
| Llama-3.1-FT | **0.565** | **0.130** | **0.018** | 0.830 | **0.120** | **0.012** | 0.602 | 0.186 | 0.193 | 0.517 | **0.161** | **0.004** | 0.577 | **0.163** | 0.077 |

open models (LLama3.1 BLEU-AVG:0.186) significantly. Within the closed models, Gemini performing better on semantic measure, in most of the regions, with GPT4o closely following. While LLama3.1 leads the open models in both the lexical and semantic measures across majority of the regions.

**High- *vs* Low-resource Languages** Figure 5 reports the average BLEU scores across all the regions, grouped by the four resource tiers: high- to extremely-low resource languages. We observed the highest performance from (L2) English and lowest for Assamese. This clearly indicates that the performance correlates to the representation and/or richness of digital content of the language used in the models.

**Fine-tuned Models** Our findings, reported in Table 3, indicates that fine-tuning with the Multi*NativQA* train set mostly improve performance for (extremely-)low resource language such as Assamese and Nepali. For the medium resources, the results are mixed. We noticed, fine-tuning helps dialect-rich language like Arabic more compare to other Mid-languages. We hypothesize, this improvement is due to the fact that fine-tuning with native dataset is enriching model's cultural (and dialectal) knowledge. For high-resource languages the performance exhibits the strength of the base model itself, as expected.

**Subjective Evaluation** We performed qualitative evaluation of GPT$-4o$ model for Assamese, Bangla_IN, and Hindi for *accuracy* and *usefulness* using a rating scale of 1-5. For the qualitative analysis, we sampled 30 QA pairs from each languages and observed an average accuracy rating of 3.57 (out of 5) and average usefulness of 3.49 (/5). Our in-depth error analysis shows that the notable errors are as follows: *(i)* inaccurate answer in case of "proper noun" related input question that seeks a specific regional-based answer (e.g., India); *(ii)* unable to answer properly in case of question, related to the current year (2024); *(iii)* inaccurate answer in case of numerical question that seeks a specific numerical value or measurement as its answer. Details examples of error analysis is in Appendix Figure 9 and 10.

## 7 CONCLUSIONS

In this paper, we propose the *NativQA* framework, to enable constructing culturally and regionally-aligned natural QA datasets with minimal human-effort. The proposed framework is scalable and language-independent, which not just facilitate creating region- and culture-based benchmarking efforts, but also resources that can be used in continual learning or fine-tuning the LLMs. We show the efficacy of the *NativQA*, by designing and developing a multilingual native QA dataset, Multi*NativQA* – from 9 regions (7 languages) encapsulating the scenario of high-low resource representation. We benchmark the Multi*NativQA* with 2 open and 2 closed LLMs. Our results indicates the superiority of closed models over open LLMs, and the performance gaps between the high to low resource languages. We observed using Multi*NativQA* dataset for fine-tuning, we can potentially inject cultural and regional knowledge in the LLMs as reflected by the performance, for e.g., of Arabic (mid-resource) and Assamese (extremely low-resource) languages. Moreover, with Multi*NativQA*, we will also release 55k additional QA pairs with no human annotation for further research.

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

# APPENDIX

## A    LIMITATIONS

The proposed framework enables the development of datasets with cultural and native information, however, it currently has several limitations. In our current study, we relied on a single search engine within the *NativQA* framework. However, this approach can be extended to include additional search engines, using a combination of engines to enrich the QA pair collection. We also relied on human-generated seed queries to collect QAs, which can be further streamlined by employing templates for languages spoken across different regions. Our study is currently limited to benchmarking various open and closed models and fine-tuning experiments with only single model. We will extend this further with more open models.

## B    ETHICS AND BROADER IMPACT

The proposed *NativQA* framework does not involve collecting any personally identifiable information. Additionally, the proposed dataset does not include any information that can offend or harm any individual, entity, organization, or society. Therefore, we do not foresee any issues that may lead to potential risks. Human annotators were paid through an external company at standard payment rates applicable to their region. Information about human annotators is not part of the dataset, and their identities remain confidential. The proposed framework and dataset will be released publicly for non-commercial research purposes. Therefore, we strongly believe that they will be beneficial for the research community.

## C    RELATED EXISTING WORK

In Table 4, we present a comparison with previous work, highlighting how the Multi*NativQA* dataset differs from prior studies.

## D    DETAILED ANNOTATION GUIDELINE

### D.1    COLLECTING SEED QUERIES

The purpose of this study is to collect natural basic QA pairs to evaluate and enhance LLMs. Our approach to collecting such QA pairs is to utilize widely used search engines with natural queries to find relevant QA pairs. We intended to find a diverse set of questions; therefore, we selected different topics as listed below.

**Topics:** Education, Travel, Events, Food and Drinks, Names and Persons, Animals, Religion, Business, Language, Sports and Games, Clothes, Tradition, Weather, Geography, General, Literature, Plants, Science, and Immigration.

For each topic, the task was to collect seed queries. While collecting the seed queries, we needed to ensure language-specific and main-city-centric information as naturally as possible, information we typically ask on search engines. For example, "Does Qatar have beaches?" or "Do I need a visa to visit Qatar?"

Table 4: The most notable existing QA datasets compared to Multi*NativQA*.

| Dataset | # of Lang | Lang | Domain | Size |
|---------|-----------|------|--------|------|
| NQ Kwiatkowski et al. (2019) | 1 | En | Wiki | 323K |
| HotpotQA Yang et al. (2018) | 1 | En | Wiki | 113K |
| TriviaQA Joshi et al. (2017) | 1 | En | Wiki, Web | 650K |
| SquAD Rajpurkar et al. (2016) | 1 | En | Wiki | 100K |
| HelpSteer Wang et al. (2023) | 1 | En | Helpfulness | 37K |
| BanglaRQA Ekram et al. (2022) | 1 | Bn | Wiki | 3k |
| TyDiQA Clark et al. (2020) | 11 | En, Ar, Bn, Fi, Id, Ja, Sw, Ko, Ru, Te, Th | Wiki | 204k |
| GooAQ Khashabi et al. (2021) | 1 | En | Open | 3M |
| BLEnD Myung et al. (2024) | 13 | En, Zh, Es, Id, Ko, El, Fa, Ar, Az, Su, As, Ha, Am | Open | 52.5k |
| CaLMQA Arora et al. (2024) | 23 | En, Ar, Zh, De, Hi, He, Hu, Ja, Ko, Es, Ru, Aa, Bal, Fo, Fj, Hil, Rn, Pap, Ps, Sm, To, Tn, Wol | Open | 1.5K |
| Multi*NativQA* dataset | 7 | Ar, As, Bn, En, Hi, Np, Tr | Open | ∼64K |

These examples are based on Qatar; however, for each language, the questions will be specific to the specified location (main city/country).

## D.2  DOMAIN RELIABILITY

For the domain reliability task annotators were tasked to review each web domain to determine its credibility and assign one of the following four reliability labels:

- **Very reliable:** The information is accepted without additional verification.
- **Partially reliable:** The information may need further verification.
- **Not sure:** Unable to verify or judge the website for any reason.
- **Completely unreliable:** The website and the information appear unreliable.

**General Characteristics**  Below are some characteristics that we have considered as criteria for a domain to be considered more reliable Schwarz & Morris (2011); Flanagin & Metzger (2007); Metzger & Flanagin (2015); Library (2010); Selejan et al. (2016).

**Overall Design:**

- The domain has a professional, polished, and attractive design. It has interactive features, is well organized, easy to navigate, loads fast, and has good response speed.
- There are no errors or broken links.
- It might have paid access to information.
- The domain name suffix is considered trustworthy (e.g., ".gov").
- Absence/limited advertising. If advertisements are present, they are good quality ads for reputable and decent products and organizations.
- The domain might be sponsored by or shows links to reputable organizations.
- Presence of a section or page on privacy and security, About page, contact info, and address.
- If videos, images, and graphics are used on the website, they are high-quality and professional.

**Content Quality:**

- Author/entity names, qualifications, credentials, and contact information are present, and they are relevant to the topic of the website or the content presented.
- Author/entity is reputable.
- Contains date stamp
- Presents information that is current and up to date.
- Has citations, especially to scientific data or references, and shows links to external authorities.
- Content is relevant to the target topic and current events.
- Professional-quality, clear writing, and good formatting of text.
- Content appears accurate, lacks bias, factually correct, plausibility, and uses appropriate objective language.
- Free of misspellings and grammar mistakes.
- The information provided is at an appropriate level, not too generic or elementary.

**General Instructions:** We also provided the following general instructions to guide annotators.

- Do not spend more than five minutes per given Web domain.
- Explore/observe/look at **ALL** elements in the domain's home page from top to bottom.
- Repeat points 1-2 on other pages from the same domain, and look at their content, structure, design, author, etc. *You are not required to read these pages in full, reading the first 1-2 paragraphs is enough.*
- During annotation, consider the annotation criteria mentioned in this guideline, and evaluate each source based on those aspects. A "reliable website" might not meet all those criteria. It is your job, as annotator, to measure the website's reliability guided by these criteria.
- You should evaluate a domain based on what is presented on it only. You should not navigate or search in outside sources, even if some are linked inside the given domain/page.
- Please use "Not sure" very sparingly in rare cases when you are extremely unsure. It is preferable to always choose one of the other three labels.
- For social media websites (e.g., X, Facebook) choose: Very Reliable.
- For shopping websites, use the criteria listed in this guideline to decide. Some shopping websites are very reliable.
- For famous people's websites, use the criteria listed in this guideline to decide.
- Websites that are in any other language ONLY (for example, only in En when you are working on Bangla queries), for such cases choose: Not Sure.

### D.3 QA ANNOTATION (DETAILED ANNOTATION GUIDELINE)

#### D.3.1 QUESTION VALIDATION:

In this task, a pair of a question and a possible answer for that question is shown. Relying only on the question shown on the interface, the annotator is asked to perform the following tasks:

1. Categorize the question as "Good" or "Bad". Steps 2- 4 will be performed only for questions labelled as "good".

2. Identify if the question is relevant to the specified location.

3. Categorize the answer.

4. Edit the answer (if needed).

The annotators classified whether the questions are "Good" or "Bad" based on the criteria discussed below. The choice of the two types of questions was inspired by the NQ dataset (Kwiatkowski et al., 2019).

- **Good question:** is a fact-seeking question that can be answered with a name of an entity (person, place, thing.etc.), or an explanation, or a number. For examples, see Table 5
- **Bad question:** A question a that meets any of the following criteria mentioned below.

Table 5: Examples of good questions in English and Arabic.

| Language | Example |
|---|---|
| EN | Is Al Wakrah Beach free? |
| | Do you have to pay for school in Qatar? |
| AR | كم اسعار الشقق في الدوحة؟ |
| | (EN Translation: How much is apartment rent in Doha?) |
| | كيف احصل على فرصة عمل في قطر؟ |
| | (EN Translation: How do I find a job opportunity in Qatar? ) |
| | كيف اقدم على وظيفة في وزارة الداخلية؟ |
| | (EN Translation: How do I apply for a job in Ministry of Foreign Affairs?) |

- It is unclear what the question is asking for. See Table 6
- Incomprehensible as a result of grammatical errors. This will be a rare case. Some grammatical mistakes can be acceptable as long as its meaning is understandable.
- The question depends on clear false presupposition, such as racist, sexist, or stereotypical ideas as shown in Table 7.
- The question is opinion-seeking. It asks the public opinion on a topic/matter where finding a concrete answer is difficult. For examples, see Table 8
- The question does not clearly ask for factual information. See Table 9

Table 6: Examples of bad questions in English, Indic Languages, and Arabic.

| Language | Example |
|---|---|
| EN | Is Indonesian coffee strong? |
| AS/BN/HI (Indic Languages) | **Note: Special cases only for India (Indic Languages)** |
| | - Who is the chief minister of India? |
| | (Reason: In India, the chief minister is referred to state-wise, for example, "Who is the chief minister of West Bengal?") |
| | - What is the national language of India? |
| | (Reason: In India, the correct question would be, "What is the official language of India?" As per the constitution, Hindi and English are the official languages, and 22 languages are recognized as per the Eighth Schedule of the Constitution. India has no national language.) |
| | - What is the national animal of Assam/West Bengal? |
| | (Reason: National animals refer to countries, not states. For example, "What is the national animal of India?") |
| AR | كيف اكتب كأس العالم؟ |
| | (EN Translation: How do I write "World Cup"?) |

## D.4 QUESTION'S RELEVANCY TO THE LOCATION

For questions labelled as "Good", the annotator is asked to identify whether the question is related to the specified [LOCATION]. Please see the examples below. For this step, one of the below labels should be chosen:

Table 7: Examples of bad questions that depends on false presuppositions in English and Arabic.

| Language | Example |
|---|---|
| EN | - What is the reason that African Americans are mostly criminals?
- Why does Islam encourage terrorism?
- Why shouldn't women be allowed to work? |
| AR | ـ لِما معظم الأفريقيين الأمريكيين مجرمين؟
(EN Translation: Why do most African American are criminals?)
ـ لماذا يشجع الإسلام على الإرهاب؟
(EN Translation: Why does Islam encourage terrorism? )
ـ لماذا يجب منع النساء من العمل؟
(EN Translation: Why women should not allowed to work? ) |

Table 8: Examples of bad questions in English and Arabic.

| Language | Example |
|---|---|
| EN | - Can you give me your thoughts on smoking?
- Is marriage good or bad? |
| AR | ـ هل من الضروري ارتداء الزي المدرسي؟
(EN Translation: Is it important to wear a school uniform?) |

Table 9: Examples of bad questions in English and Arabic.

| Language | Example |
|---|---|
| EN | - How do you ensure you are culturally competent?
- Why is it a must to preserve our local literature? |
| AR | ـ هل من السهل ايجاد عمل في قطر؟
(EN Translation: Is it easy to find job in Qatar? )
ـ كم يستغرق الطلب تحت الاجراء قطر؟
(EN Translation: How long does "in process" take Qatar? ) |

- **Yes:** The question specifically relates to the location. For examples, see Table 10

- **No:** The question is not related to the specified location, but could be related to a different location. See Table 11

- **Maybe:** The question is somewhat generic. It could apply to the specified location, but it might also be relevant to other locations. For examples, see Table 12

- **Unsure:** It's challenging to determine if the question is location-specific. This option should be chosen only for particularly difficult cases. For examples, see Table 13

Table 10: Examples of questions in English and Arabic.

| Language | Example |
| --- | --- |
| EN | What is the main city in Qatar? |
| AR | هل قطر لديها ملك؟ |
| | **Translation:** Does Qatar have a king? |
| | كم عدد المساجد في دولة قطر؟ |
| | **Translation:** How many mosques are there in Qatar? |

Table 11: Examples of questions in English and Arabic with specific locations.

| Language | Example |
| --- | --- |
| EN | Why do Emirati men wear white robes? (the specific location was Qatar) |
| AR | ما هي اقامة مستثمر في السعودية؟ |
| | **Translation:** What is investor residency is Saudi Arabia? |
| | (الموقع المطلوب كان قطر) |
| | **Translation:** The specified location in Qatar. |

Table 12: Examples of generic questions in English and Arabic.

| Language | Example |
| --- | --- |
| EN | - What is the most visited mall? 
 - What is a place where bread and cakes are sold? |
| AR | - كم عدد كليات الطب؟ |
| | **Translation:** How many medical colleges? |
| | - كم الدرجة المطلوبة في اختبار الايلتس؟ |
| | **Translation:** What is the required grade for ILETS? |

## D.5 ANSWER CATEGORIZATION:

The answer of the given question should be classified using one of the below categories. The source Web page provided on the interface should be used to make the judgment.

- **Correct answer:** When the answer aligns with the information provided by the source. Note that the answer must be complete and addresses all parts of the question, but it does not need to match the source webpage verbatim. The answer can be a long, detailed response, or a short snippet.

Table 13: Examples of questions in English and Arabic.

| Language | Example |
|---|---|
| **EN** | - Is DoorDash cheaper or Uber Eats?
- What are common names for Paspalum? |
| **AR** | ـ كيف تعرف الصقر وهو في الجو؟

**Translation:** How to know the falcon while he is in the air?

ـ ما معنى اسم عطشان؟

**Translation:** What is the meaning of the name "Thirsty"? |

- **Partially correct answer:** When the answer does not address all parts of the question. In this case, the answer should be edited using information from the source page. The required information can be directly copied from the source webpage. Minimal editing may be needed to make the answer more comprehensive. For example, see Table 14

- **Incorrect answer:** When the answer does not address the question at all. In this case, the answer should be edited using information from the source page. See Table15

- **Cannot find answer:** When the answer is not available in the provided link/page, and thus, cannot be judged.

Table 14: Examples of questions and answers in English and Arabic. The answers provide more information and should be edited.

| Language | Question | Answer |
|---|---|---|
| **EN** | How many Americans live in Qatar? | In recent years, this figure has more than doubled and various estimates now put the number of Americans in Qatar to be up to 15,000. Most Americans within the country tend to be based in the capital city of Doha and are largely attracted by the tax-free inducement of the Persian Gulf state. |
| **AR** | من أكبر البحرين أو قطر؟

(EN Translation: Which is bigger: Bahrain or Qatar? ) | تتنوع مساحة الدول العربية بشكل كبير، حيث تبلغ مساحة أكبر دولة عربية، وهي الجزائر، ٢٠٣٨١٠٧٤١ كيلومتر مربع، بينما تبلغ مساحة أصغر دولة عربية، وهي البحرين، ٧٨٥ كيلومتر مربع، وفقا لآخر تحديث لموقع worldometers.
**Translation:** The area of the Arab countries varies greatly, as the area of the largest Arab country, Algeria, is 2,381,741 square kilometers, while the area of the smallest Arab country, Bahrain, is 785 square kilometers, according to the latest update to the website Worldometers. |

**Answer editing:** for the cases that require the answers to be edited, the below instructions should be followed:

- The parts that completely answers the question should be copied from the webpage and pasted in the answer box on the interface. This could be a long paragraph or a short snippet, or runs through multiple paragraphs.

- Sometimes answers may end with: (. . . ), in such cases, the answer should be completed by finding the remaining part of the answers in the webpage.

- The answer should be to the point and concise. For example, if the question asks for the colour of a flag, then the answer should only answer that. Any unnecessary parts should be removed.

Table 15: Examples of questions and wrong answers in English and Arabic. The answers need to be edited.

| Language | Question | Answer |
|---|---|---|
| EN | Does Qatar have online shopping? | Carrefour Qatar - Shop Online for Grocery, Food, Mobiles, Electronics, Beauty, Baby Care & More. |
| AR | من هي اغنى عائلة في قطر؟ 

 **Translation:** Who is the richest family in Qatar? | جاءت عائلة ساويرس في المرتبة الأولى كأغنى عائلة في المنطقة العربية، بصافي ثروة إجمالية قدرها ٢٠١١ مليار دولار. 

 **Translation:** The Sawiris family ranked first as the richest family in the Arab region, with a total net worth of 11.2 billion dollar. |

## D.6 Annotation Platform

We utilized in-house annotation platform for the tasks. Separate annotation interfaces (as presented in Appendix K) were designed for each phase and each language, resulting 18 annotation projects. To facilitate the annotation process, the annotation interface included the annotation guidelines throughout the phases.

## E Prompting and Instruction Tuning: Additional Details

### E.1 Prompts

In our main experiments of zero-shot prompting of the different LLMs, we manually and carefully designed a prompt to instruct a model to perform the QA task. Our prompt engineering process is inspired by relevant research and our experimental observations over the development sets. For this experiment, we use the system and user prompts in Table 16.

Table 16: Prompts used with the LLMs for zero-shot question answering. *lang*: the language of QA pair.

| Role | Prompt |
|---|---|
| System | You are a/an **[*lang*]** AI assistant specializing in both short and long-form question answering. Your task is to provide clear, accurate, and relevant responses across various fields, ensuring concise and well-structured answers. |
| User | Please use your expertise to answer the following **[*lang*]** question. Answer in **[*lang*]** and rate your confidence level from 1 to 10. Provide your response in the following JSON format: {"answer": "your answer", "score": your confidence score}. Please provide JSON output only. No additional text. **Question**: input_question |

### E.2 Instruction Generation

To generate instruction templates through GPT-4o and Claude-3.5 Sonnet, we use the prompt in Table 17. Table 18 shows examples of the generated instructions. Note that we only generate instructions for the user role, while we keep the system role fixed to that presented in Table 18. For all generated instructions, we append the following suffix to the instruction to further instruct the LLM to comply to our requirement of concise answers: *Make your answer very concise and to the point. Return only the answer without any explanation, justification or additional text.*

Table 17: Prompts used to generate instructions through LLMs.

| Role | Prompt |
|---|---|
| System | You are an expert LLM developer with expertise in writing instructions to instruction-tune LLMs for users' tasks. |
| User | We are creating an English instruction-following dataset for question answering task. An example instruction is: Interpret the following question about the real world carefully and research each answer, then provide a clear and concise answer to the question. Write 10 very diverse and concise English instructions. Only return the instructions without additional text. Return the instructions as strings in a list format as follows: [] |

| Model | Instruction | System Role |
|---|---|---|
| GPT-4o | Analyze the given question thoroughly and provide a well-researched and precise answer. | You are a/an [*lang*] AI assistant specialized in providing detailed and accurate answers across various fields. Your task is to deliver clear, concise, and relevant information. |
| Claude-1.5 | Carefully consider the question and provide a short, well-researched answer that covers all key points. | You are a/an [*lang*] AI assistant specialized in providing detailed and accurate answers across various fields. Your task is to deliver clear, concise, and relevant information. |

Table 18: Examples of instructions generated by two LLMs along with the pre-defined system role prompt. *lang*: the language of QA pairs for which the final instruction will be created.

## F    DATASET: ADDITIONAL DATA

In addition to the dataset summarized in Table 1, we have collected un-annotated QA pairs for additional locations. Table 19 shows statistics of collected Arabic and English data in different locations.

Table 19: Statistics of additional QA pairs collected for different locations through our framework.

| Lang-Loc | # of QA | Lang-Loc | # of QA |
|---|---|---|---|
| Arabic-Egypt | 7,956 | Arabic-Tunisia | 14,789 |
| Arabic-Palestine | 5,679 | Arabic-Yemen | 4,818 |
| Arabic-Sudan | 4,718 | English-New York | 6,454 |
| **Total** | | | **55,702** |

## G    ANNOTATED DATASET: ADDITIONAL DETAILS

In Figure 6, 7 and 8 we present the topic-wise data distribution for different datasets associated with various languages. Starting with the Arabic dataset, the predominant topic is *names*, comprising 10.6% of the data. For Assamese, the major category is Literature (14.6%). For Bangla, whether from Bangladesh or India, the major topic is *general*, representing 8.8% and 9.8% respectively. In Bangladesh, *religion* (10.7%) is the major topic for English, whereas in Qatar, *general* dominates at 26.5% and Food and drinks dominates a

second major topic. For Nepali, the leading topic is *General* (19.8%), for Hindi it is *Travel* and Plant (8.1% for each topic), and for Turkish, *names* is the primary topic at 8.7%.

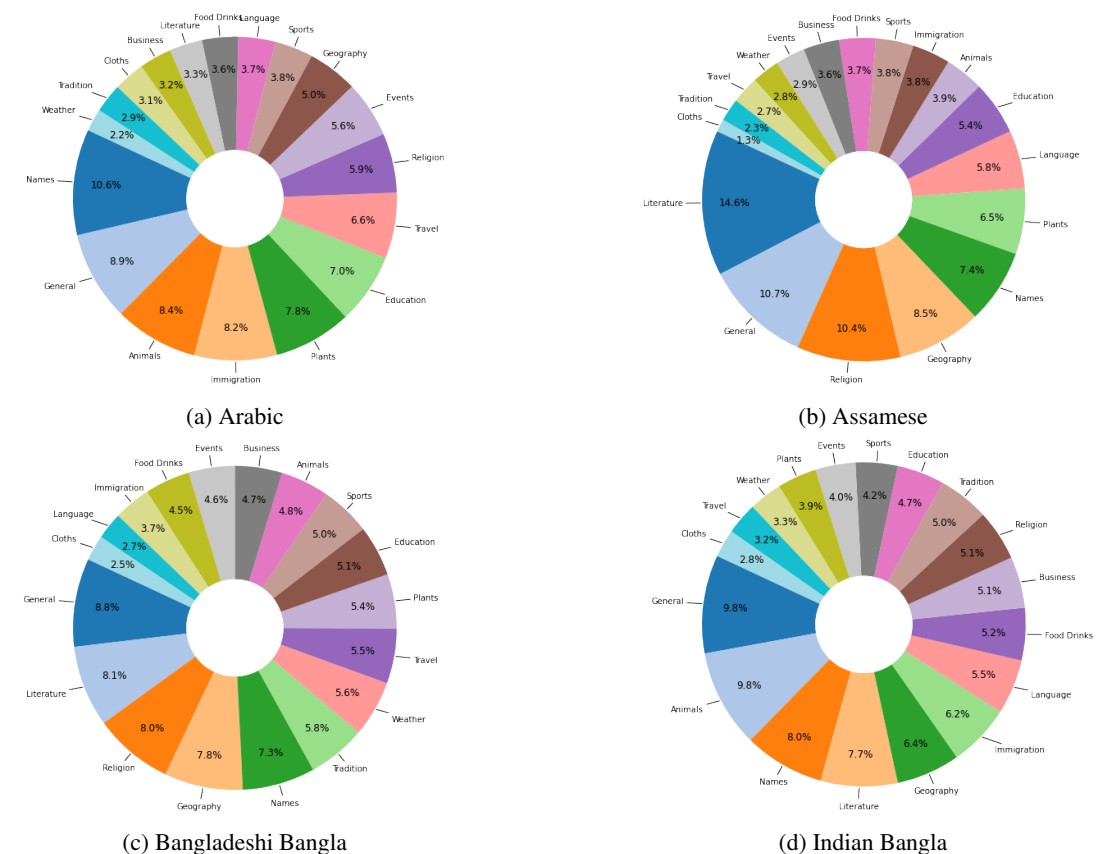

Figure 6: Topic wise distribution in different languages such as *Arabic*, *Assamese*, *Bangladeshi Bangla*, and *Indian Bangla*,

# H DATASET: ANNOTATION (ANSWER EDITING) ANALYSIS

We computed the normalized Levenshtein distance between the original answer collected using *NativQA* framework and the annotated answer to identify the robustness of *NativQA* framework. During the distance computation, we provide a weight of 1 for insertion, deletion, and substitution operations. The average edits across all languages are relatively low (0.17), which indicates minimal edits has been done on the answers. In Table 20, we provide distance measures for all languages across different data splits. As shown in the table, the majority of edits were made for Hindi, Nepali, and Bangla (IN), with distance measures of 0.336, 0.302, and 0.266, respectively. Overall, the edits are relatively low across languages, suggesting that the semi-supervised approach used in the *NativQA* framework can be adapted for creating resources for other languages and locations.

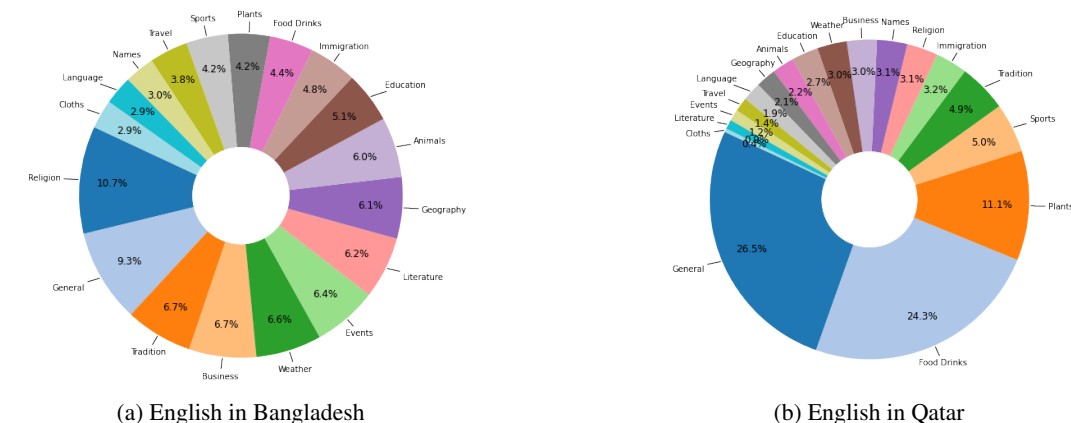

(a) English in Bangladesh       (b) English in Qatar

Figure 7: Topic wise distribution in different languages such as *English in Bangladesh*, and *English in Qatar*.

Table 20: Normalized Levenshtein distance for all languages across different splits. *Average (Split)* indicates on average distance measure across splits. −− No training and dev sets for Nepali.

| Data Split | Arabic | Assamese | Bangla (BD) | Bangla (IN) | English (BD) |
|---|---|---|---|---|---|
| Train | 0.196 | 0.136 | 0.191 | 0.265 | 0.114 |
| Dev | 0.063 | 0.096 | 0.307 | 0.366 | 0.160 |
| Test | 0.229 | 0.165 | 0.005 | 0.166 | 0.001 |
| **Average** | 0.163 | 0.132 | 0.168 | 0.266 | 0.092 |
| | **English (QA)** | **Hindi** | **Nepali** | **Turkish** | **Average (Split)** |
| Train | 0.149 | 0.362 | - | 0.052 | 0.188 |
| Dev | 0.053 | 0.186 | - | 0.190 | 0.143 |
| Test | 0.043 | 0.460 | 0.302 | 0.186 | 0.248 |
| **Average** | 0.082 | 0.336 | 0.302 | 0.143 | |

## I  LANGUAGE SPECIFIC MODELS FOR BERTSCORE

In Table 21, we present the pre-trained language models used with BERTScore to account for language-specific variations in the evaluation measures.

## J  DETAILS OF THE ERROR ANALYSIS (SUBJECTIVE EVALUATION)

The goal of the human evaluation task was to rate the *accuracy* and *usefulness* of an LLM's output. The rating scale ranges from 1 to 5, where higher values indicate better performance in both categories. We defined the measures and their guidelines as follows:

**Accuracy:** Measures whether the answer is factually correct and aligns with established knowledge or the provided context. Consider whether the answer presented is free from errors, consistent with known information, and precise in its claims. The rating score representing accuracy is as follows:

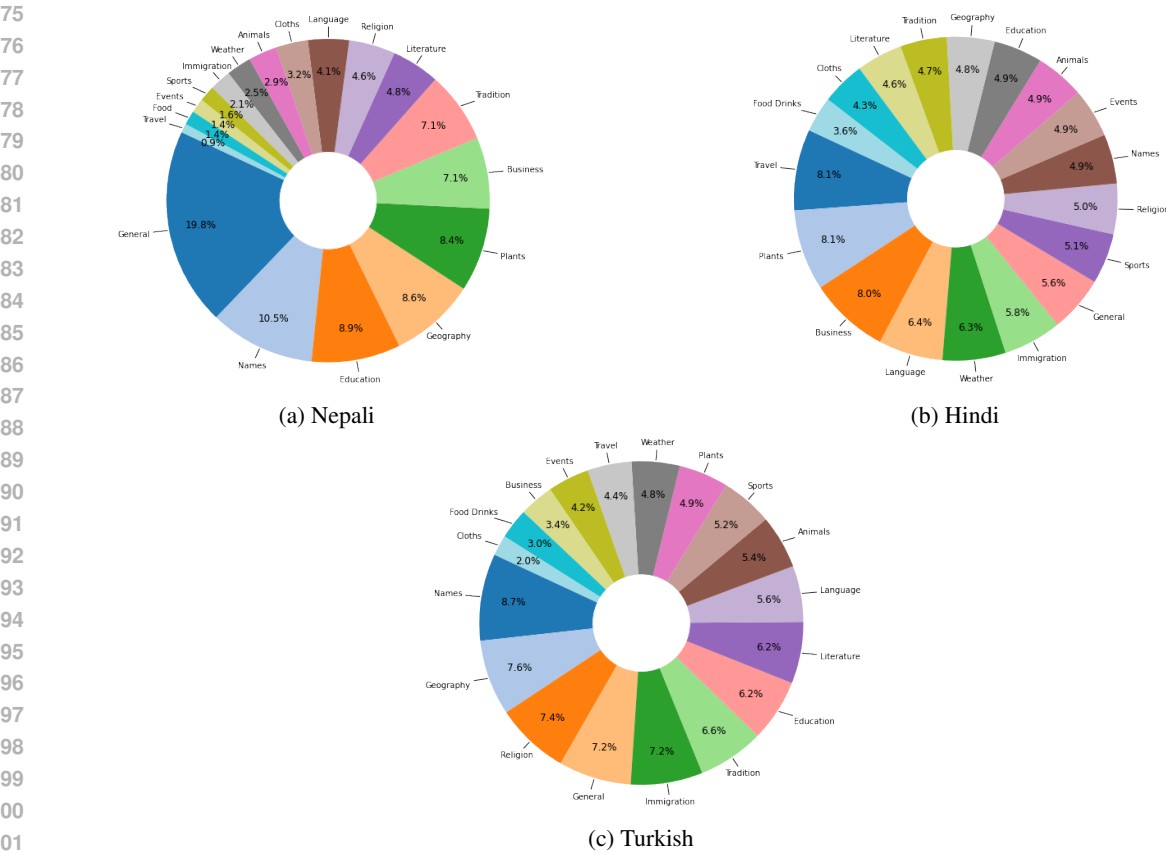

Figure 8: Topic wise distribution in different languages such as *Nepali*, *Hindi* and *Turkish*.

Table 21: Language specific models used to compute BERTSCore. Model id is same on HuggingFace.

| Language/Region | Model |
|---|---|
| Arabic | aubmindlab/bert-base-arabertv2 |
| Assamese | ai4bharat/indic-bert |
| Bangla (BD) | csebuetnlp/banglabert |
| Bangla (IN) | sagorsarker/bangla-bert-base |
| English (BD) | bert-base-uncased |
| English (QA) | bert-base-uncased |
| Hindi | ai4bharat/indic-bert |
| Nepali | bert-base-multilingual-uncased |
| Turkish | dbmdz/bert-base-turkish-cased |

*5: Very Accurate* The answer is completely accurate, without any errors. All claims and facts presented are correct and aligned with the expected answer. There is no misleading or incorrect information.

*4: Accurate* The answer is mostly accurate, with only minor or negligible inaccuracies. There may be small factual inconsistencies that do not significantly affect the overall meaning or quality of the answer.

*3: Neutral (neither accurate nor inaccurate)* The answer is somewhat accurate but also contains elements of inaccuracy. It is neither highly accurate nor does it contain substantial errors.

*2: Inaccurate* The answer contains multiple factual errors or inaccuracies that detract from its overall quality. While the core meaning might still be understandable, important details are incorrect or misleading.

*1: Very Inaccurate* The answer is largely or completely inaccurate. It does not align with the expected or correct information.

**Usefulness:** It evaluates how helpful, relevant, and applicable the answer is for addressing the task or question at hand. The rating score representing usefulness is as follows:

*5: Very Useful* The answer is highly useful and provides all necessary information in a clear, and concise manner.

*4: Useful* The answer is useful but may not be exhaustive. It provides relevant information for which question is asked.

*3: Neutral (neither useful nor not useful)* The answer is somewhat useful but lacks all information.

*2: Slightly Useful* The answer is minimally useful, offering less information. The overall output does not sufficiently answer the question.

*1: Not Useful at All* The answer is completely unhelpful and irrelevant.

**Human (Subjective) Evaluation:** We randomly sampled 30 QA pairs with answers generated by GPT-4 from a dataset in Assamese, Bangla-IN, and Hindi for human evaluation. Due to limited resources, we were unable to extend the evaluation to other languages. Following the definitions and instructions provided above, human evaluators scored the answers. For accuracy, the scores were 4.03 for Assamese, 3.13 for Bangla-IN, and 3.56 for Hindi. For usefulness, the scores were 4.00 for Assamese, 3.06 for Bangla-IN, and 3.43 for Hindi. In Figure 9 and 10 we report samples of QA pairs for Assamese, Bangla (IN), and Hindi, demonstrating the answer from GPT-4o and reference. Also, it is observed that the GPT$-4$ answer is short while the reference answer is long. However, it is the opposite in other cases, which impacts the overall performance measures.

## K    ANNOTATION INTERFACE

In Figure 11, we present a screenshot of the interface designed for domain reliability check, which consisted of a URL of the domain, annotation guidelines, and four different options associated with the four categories we defined for this annotation task. Annotators select one of these labels and submit.

In Figure 12 and 13 we provide a screenshot of the interface that demonstrate the steps of question validation, question's relevancy to the location, answer categorization and editing the answer, respectively. The later steps will appear on the interface depending on the classification of the question in question validation step.

## L    DATA RELEASE AND LICENSE

The *NativQA* dataset will be publicly released under the Creative Commons Attribution Non Commercial Share Alike 4.0: `https://creativecommons.org/licenses/by-nc-sa/4.0/`.

| Question | GPT-4o (Answer) | Reference |
|---|---|---|
| **Assamese** | | |
| বাক্সা জিলাৰ বৰ্তমান উপায়ুক্ত গৰাকীৰ নাম কি ? | বিজয় কৃষ্ণ কুমাৰ | বাক্সা জিলাৰ বৰ্তমান উপায়ুক্ত ৰঞ্জন শৰ্মা (২০২০ চন)। |
| What is the name of the current Commissioner of Baxa District? | Vijay Krishna Kumar | The present Commissioner of Baxa District is Ranjan Sharma (2020). |
| অসম সাহিত্য সভাৰ প্ৰধান সম্পাদক জনৰ নাম কি ? | অসম সাহিত্য সভাৰ নতুন সভাপতি ড· সূৰ্য্যকান্ত হাজৰিকা- প্ৰধান সম্পাদক ড· উপেন্দ্ৰজিৎ শৰ্মা - নিয়মীয়া বাৰ্তা | ৰামচৰণ ঠাকুৰ |
| What is the name of the Secretary General of the Assam Sahitya Sabha? | Dr. Suryakanta Hazarika, New President of Asom Sahitya Sabha - Editor-in-Chief Dr. Upendrajit Sharma - Regular Message | Ramcharan Tagore |
| **Bangla (IN)** | | |
| সুন্দরবন অঞ্চলে নতুন দ্বীপটির নাম কি ? | নতুন দ্বীপটির নাম জানা নেই | ঘোড়ামারা দ্বীপ হল ভারতের পশ্চিমবঙ্গ রাজ্যের সুন্দরবন এলাকার একটি দ্বীপ। |
| What is the name of the new island in the Sundarbans region? | The name of the new island is unknown | Ghoramara Island is an island in the Sundarbans region of the Indian state of West Bengal. |
| সরিষার তেল কোনটি 100% খাঁটি ? | ব্র্যান্ড নির্ভর করে খাঁটি সরিষার তেল নির্ধারণ করা যায়। | জাজা হলুদ সরিষার তেল হল 100% বিশুদ্ধ এবং প্রাকৃতিক কোল্ড প্রেসড তেল। |
| Which mustard oil is 100% pure? | Pure mustard oil can be determined depending on the brand. | Jaja Yellow Mustard Oil is 100% pure and natural cold pressed oil. |

Figure 9: QA pairs with GPT-4o answer and reference for *Assamese* and *Bangla-IN* (with English translation), highlighting potential errors.

| Question | (Answer) | Reference |
|---|---|---|
| **Hindi** | | |
| क्या 2024 में मानसून ने केरल में दस्तक दी थी? | मुझे खेद है, लेकिन मेरे पास 2024 के मानसून के बारे में जानकारी नहीं है। | भारत मौसम विज्ञान विभाग (आईएमडी) के एक बयान में गुरुवार को कहा गया कि दक्षिण–पश्चिम मानसून सामान्य से दो दिन पहले और पूर्वानुमानित तारीख 31 मई से एक दिन पहले गुरुवार को केरल में पहुंच गया । एक असामान्य मोड़ में, मानसून भी एक साथ पूर्वोत्तर भारत में प्रवेश कर गया। |
| Did monsoon hit Kerala in 2024? | I am sorry, but I do not have information about monsoon 2024. | The southwest monsoon reached Kerala on Thursday, two days earlier than normal and a day before the predicted date of May 31, an India Meteorological Department (IMD) statement said on Thursday. In an unusual twist, the monsoon also entered Northeast India simultaneously. |
| तीनों सेना के अध्यक्ष कौन है 2024? | मुझे खेद है, लेकिन 2024 के लिए तीनों सेना के अध्यक्ष की जानकारी मेरे पास नहीं है। | फिलहाल भारतीय सेना के चीफ ऑफ आर्मी स्टाफ हैं जनरल मनोज पांडे. साल 2022 में पूर्व जनरल मनोज मुकुंद नरवणे के रिटायर होने के बाद जनरल मनोज पांडे ने यह पद संभाला था। |
| Who is the Chief of the three Services 2024? | I am sorry, but I do not have the information of the chairman of the three services for 2024. | At present, General Manoj Pandey is the Chief of Army Staff of the Indian Army. After the retirement of former General Manoj Mukund Naravane in the year 2022, General Manoj Pandey took over this post. |

Figure 10: QA pairs with GPT-4o answer and reference for *Hindi* (with English translation), highlighting potential errors.

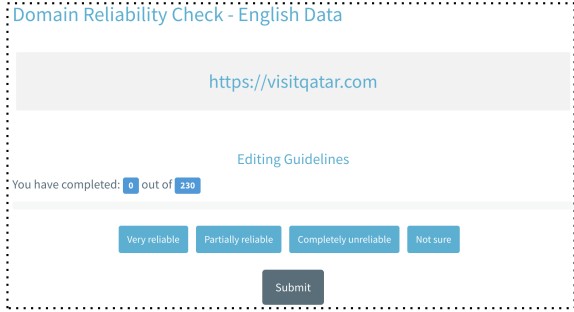

Figure 11: An example of the annotation interface for domain reliability check.

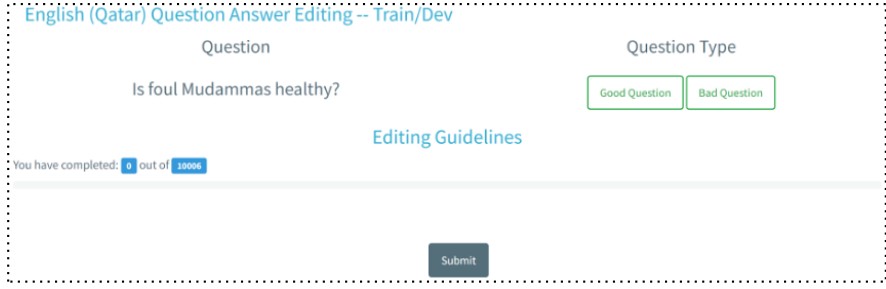

Figure 12: Annotation interface for *Question Validation*.

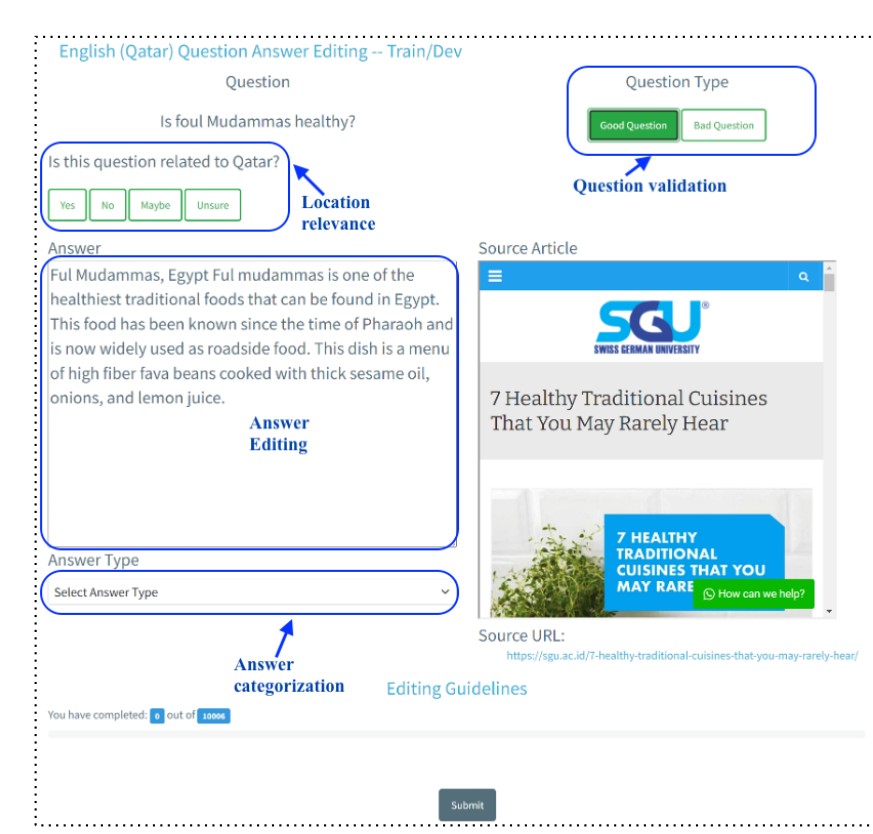

Figure 13: Annotation interface for *question validation*, *location relevance*, *answer editing*, and *answer categorization*.

