# OpenReview forum: "NativQA: Multilingual Culturally-Aligned Natural Queries for LLMs"
_ICLR.cc/2025/Conference — Submitted to ICLR 2025_

### Official Review · Reviewer_mhhX · 2024-10-30

**Soundness:** 2
**Presentation:** 3
**Contribution:** 2
**Rating:** 8
**Confidence:** 4

**Summary:**

The authors present a strategy to scalably collect QA pairs for lower resource languages, by leveraging Google’s “People also asked” feature. MultiNativQA spans 64k QA pairs that undergo manual verification, and are used as a resource for culturally-relevant training and evaluation.

**Strengths:**

* The paper is well written and has clear contributions—especially their method of scalably collecting QA pairs, by leveraging Google’s “People also asked” feature.
* The authors take care to verify the correctness of the domains, the questions, and the answers, in a multi-stage verification process. This attention to detail is notable.
* The authors address a key problem in collecting better training and evaluation data for languages that have less resources.
* The manual qualitative analysis of errors from major models is quite useful, and an important set of insights.

**Weaknesses:**

* Invariably, the naturalness of the questions may be skewed by the seed set, which is constructed by annotators who are asked to think of natural questions for their region. To some extent this is artificial, but also a difficult hurdle to circumvent. The authors could strengthen the work by listing how many seed authors there were for each language, to illustrate the extent that there was diversity of thinking.
* It isn’t clear the extent to which the information in the dataset is novel to the tested models, because BERT-Score, ROUGE, and BLEU are fairly unreliable scores, especially multilingually. It would have been stronger to conduct human reviews of the answers and calculate the correlations with automatic scores, to give an idea of whether they are reliable.

**Questions:**

* Why only deduplicate synthetic queries with exact matches?
* Was Google always accessed from the region in question or using VPN, to ensure geographic consistency?
* Many of the questions in your example tables are time sensitive. Do you consider keeping time stamps with all these questions, in case the answers change?

---

### Official Review · Reviewer_LQGG · 2024-11-03

**Soundness:** 3
**Presentation:** 3
**Contribution:** 3
**Rating:** 8
**Confidence:** 5

**Summary:**

The paper introduces NativQA, a framework designed to create culturally and regionally aligned multilingual QA datasets for large language models (LLMs), addressing the lack of region-specific datasets generated by native users in their languages. This gap has limited the effectiveness of LLMs in real-world applications, particularly in low-resource and culturally rich languages. The framework includes a scalable, language-independent approach that combines human and machine collaboration for QA collection, enabling minimal human intervention. Key contributions of the paper are:

- NativQA Framework: A semi-automatic approach to build culturally specific QA datasets that enhances LLM inclusivity.
- MultiNativQA Dataset: A multilingual QA dataset with approximately 64,000 annotated pairs across seven languages, representing various resources (high to extremely low) and covering 18 topics from nine regions.
- Benchmarking: Evaluation of both open and closed-source LLMs on the MultiNativQA dataset.
- Parameter-Efficient Fine-Tuning: The use of PEFT techniques demonstrated the dataset's efficacy for enhancing LLMs' cultural and regional awareness.

The authors highlight that the dataset and framework fill a significant gap in creating LLM benchmarks that align with diverse linguistic and cultural contexts​

**Strengths:**

Originality

Through NativQA, the authors address a previously underexplored area: the creation of QA datasets that are culturally and regionally relevant, especially for languages with limited digital resources. This contribution moves beyond standard methods that rely on translations from English datasets, which often miss the nuanced needs of native speakers and culturally specific knowledge. The integration of a semi-automated process combining native query input with machine-assisted QA generation is particularly novel, reducing the dependency on purely manual or translation-based methods. This approach not only mitigates costs but also opens up new avenues for evaluating and fine-tuning LLMs in low-resource, culturally rich settings, setting it apart from existing QA datasets.

Quality

The quality of this work is supported by a rigorous, three-part framework, which includes query collection, QA pair extraction, and quality validation. The authors demonstrate a well-structured and scalable pipeline that ensures the QA data is both culturally relevant and diverse. The design of the MultiNativQA dataset, with over 64,000 annotated pairs across seven languages and nine regions, provides robust benchmarking capabilities. The authors also benchmark the performance of both open and closed-source LLMs on this dataset, offering a thorough evaluation. Additionally, the paper's use of parameter-efficient fine-tuning (PEFT) is well-justified and strategically chosen to address computational costs, a thoughtful addition to the framework's design.

Clarity

The paper is largely clear and methodical in its presentation. The breakdown of the framework’s three modules—Query Collection, QA Collection, and QA Validation—offers a logical flow and helps readers understand each step of the process. The choice of examples from different languages further elucidates the framework’s application across diverse contexts. Tables and figures, such as those illustrating dataset distribution and evaluation metrics, are well-designed and aid in comprehending the results. However, a more detailed discussion of the potential limitations in the methodology (e.g., reliance on a single search engine) could further improve clarity in terms of the framework’s scope.

Significance

The significance of NativQA is considerable, both in terms of immediate applications and future impact. Culturally and regionally relevant datasets are essential for deploying LLMs in real-world applications that involve diverse linguistic groups. By making both the NativQA framework and the MultiNativQA dataset publicly available, the authors enable the broader NLP community to explore and develop culturally aware models, especially for underrepresented languages. This dataset can potentially become a standard for future evaluations of LLMs on low-resource languages, encouraging the development of models that are better aligned with regional and cultural nuances. Additionally, the fine-tuning experiments with PEFT demonstrate a practical, resource-efficient method for enhancing LLMs with the MultiNativQA data, suggesting paths for further research in adaptive fine-tuning strategies.

**Weaknesses:**

1. Limited Scope in Query Collection Sources
Weakness: The paper relies heavily on a single search engine (Google) for QA collection, which may not always yield the culturally diverse content required for all languages and regions, particularly for low-resource languages. This reliance introduces potential biases, as Google’s indexing and search result prioritization may skew towards more widely available or commercial content, possibly missing nuanced, locally relevant knowledge.

Recommendation: Expanding the query sources to include regional search engines, local knowledge databases, or even social media platforms specific to target cultures would enhance the cultural alignment of the collected QA pairs. Alternatively, a comparison of QA pairs from different sources could strengthen the dataset’s diversity and reduce source bias, offering insights into the efficacy of different sources for culturally rich QA data.

2. Inconsistent QA Pair Quality in Low-Resource Languages
Weakness: The quality of QA pairs in extremely low-resource languages like Assamese and Nepali showed significant variability, with a notable decrease in usable QA pairs post-annotation. This limitation could reflect inherent challenges in capturing native content or the translation issues arising when the search engine retrieves results in more dominant languages.

Recommendation: To address this, a two-step QA collection approach could be beneficial: first, gather region-specific web content from trusted local sources in each language, then apply language detection tools to filter out non-native responses effectively. The use of pre-trained language-specific filters or bilingual annotators could improve the consistency and relevance of collected QAs, especially in low-resource settings where non-native responses are common.

3. Narrow Evaluation of Fine-Tuning Efficacy
Weakness: The fine-tuning evaluation, while demonstrating the dataset’s utility, is conducted on a single LLM (Llama-3.1-8B-Instruct) using parameter-efficient fine-tuning (PEFT) techniques. This limited evaluation scope might not provide a comprehensive view of how effectively MultiNativQA can adapt models of varying architectures and training paradigms to cultural contexts. For instance, closed-source models could behave differently compared to open-source models after fine-tuning.

Recommendation: Extending the fine-tuning experiments to a broader set of LLMs, including both transformer-based and non-transformer architectures, would enhance the robustness of the findings. Alternatively, conducting ablation studies that isolate the effects of PEFT could offer insights into the dataset’s specific impact on language and cultural alignment. If computational costs are prohibitive, an experiment with a smaller subset of the dataset across diverse models could still provide valuable insights.

4. Absence of Qualitative Cultural Alignment Analysis
Weakness: Although quantitative metrics (e.g., BLEU, ROUGE, F1 BERTScore) are valuable for measuring accuracy and fluency, they fall short in assessing how well the fine-tuned models respond to culturally nuanced or sensitive queries. An evaluation based solely on standard metrics might miss subtle cultural misalignments in the generated responses.

Recommendation: Incorporating a qualitative analysis where native speakers or cultural experts evaluate the responses for cultural appropriateness, sensitivity, and relevance could offer an additional layer of validation. This approach would complement the quantitative results, particularly for languages with rich cultural nuances that are difficult to measure with traditional metrics. An example might include a sample of questions from the MultiNativQA dataset, evaluated through a Likert scale for cultural alignment, with native speakers providing specific feedback on cultural appropriateness.

5. Ambiguity in Defining “Reliable” Sources for QA Pairs
Weakness: The paper mentions a domain reliability check, but it does not provide detailed criteria for how reliability was assessed or examples of sources that were classified under each reliability label. This lack of transparency could lead to inconsistency in determining the factuality of answers, especially for culturally specific knowledge, where conventional definitions of reliability may not apply.

Recommendation: Providing a clearer definition of reliability criteria and examples for each category (e.g., “very reliable,” “partially reliable”) would enhance reproducibility and make it easier to apply the framework to new languages and cultures. Additionally, discussing specific challenges or biases encountered in assessing domain reliability in low-resource languages would add depth to the paper’s contributions and offer a guide for future researchers working with culturally sensitive data.

6. Absence of Error Analysis for Model Limitations on Low-Resource Languages
Weakness: While the paper mentions qualitative error types (e.g., numerical errors, temporal inaccuracies), it lacks an in-depth error analysis that connects these errors with the limitations of the dataset or fine-tuning approach, especially for low-resource languages. Understanding where the models fall short could help guide future iterations of the dataset or model improvements.

Recommendation: A detailed error analysis that categorizes mistakes by language, cultural content, or query type (e.g., temporal questions, region-specific facts) would provide actionable insights. For instance, if numerical inaccuracy is prominent in certain language pairs, the analysis could propose dataset improvements or filtering strategies for factual consistency. This analysis could also help identify specific areas where the fine-tuned model struggles, offering valuable direction for future research in adapting LLMs to culturally specific information.

**Questions:**

N/A

---

### Official Review · Reviewer_mGqr · 2024-11-04

**Soundness:** 3
**Presentation:** 2
**Contribution:** 2
**Rating:** 3
**Confidence:** 4

**Summary:**

The paper presents a pipeline for collecting QA data, that leverages functions and outputs from Google Search (find relevant questions, providing answer to the questions) that can be applied to generate questions in a wide range languages. Then, they manually filter the generated QA pairs. Their new pipeline enables large-scale data collection. Paired with this framework, they also present 64K QA datasets on 7 languages.

They evaluate LLM's performance on their dataset mainly with lexical based metrics (F1, BLEU, Rouge), showing that (1) performance drops as we move from high-resource to low-resource languages, and (2) the performance of closed-sourced models (GPT-4o and Gemini) outperforms that of open-sourced models (Llama-3.1, Mistral).

While multilingual dataset, and question answering dataset, can be helpful for ICLR community, I find it hard to recommend the paper in its current form. The novelty from existing datasets is unclear, documentation of dataset is lacking, and I'm concerned about the quality of the dataset as well. Please see weaknesses section for details.

**Strengths:**

The paper presents a useful artifact, MultiNativQA dataset, that contains QA pairs in 7 languages, totaling 64K question answering datasets. The paper proposes human-in-the-loop data collection approach that leverages existing tools (e.g., Google's search outputs) smartly.
The paper is easy to follow and writing is relatively clear.

**Weaknesses:**

The contribution is not substantial enough. For a dataset paper / dataset generating framework paper, I think few key questions are:

These questions are related to the novelty of the dataset:
* What are new capabilities of models that we weren't able to evaluate before that we can now evaluate with this benchmark? How does the new dataset qualitatively or quantitatively differ from existing datasets?
  - More comparison with existing work would strengthen this work. Beyond the scale, how does the dataset differ from CaLMQA (Arora et al 24)?
* What do we learn about models through this dataset?
   - The result discussion in section 6 can be improved. How does the findings in this paper differ from other findings in the prior work? contextualizing will be helpful.

The data collection should be documented more carefully.
* The annotator statistics (e.g., how many annotators per language, how they are compensated, etc).
* Data collection cost: what's the estimated cost for collecting dataset through this framework on other languages, and how much did it cost to collect MultiNativQA? Is this framework viable for other researchers without access to your inhouse annotator pool?
* Data statistics: How long are the queries and answers typically?
* Over how many examples did you compute inter-annotator agreement? If on a subset, how did you select a subset?

The validity of the dataset/evaluation quality:
* I think key concern that I have is the quality of the answer that is automatically extracted and included in this dataset. The paper presents filtering step (Section 3.3) and agreement statistics (4.4) but it's not very convicing. What does is the matching of 66.04% mean in answer editing task? Can we compare this to some existing datasets?
* Is lexical based evaluation metric sufficient for this dataset?

I find the introduction to be vague. It’d be good to summarize concrete experimental findings (e.g., what are the result from benchmarking 2 open and 2 closed LLMs, as well as the fine-tuned LLM)?

**Questions:**

* Why is the inter annotator agreement for English (line 347) so low? Any justification for it?
* Figure 1: caption can be a bit more informative. Where are these examples from?
* How do you decide whether a language is low resource or high resource? There can be multiple definitions, it’d be good to clear how you define “extremely low” to “high” resource language, that you argue as included in the part of the dataset.
* How do you define “domain”? The domain reliability check part should be written a bit more clearly. How many domains were manually checked, and what is the rate of untrustworthy domains?
* Providing example instances from the dataset across the wide range of languages (with gloss) in the appendix will be helpful.

Minor comments:
Table 5,...13.. - the Arabic questions should come with English gloss for readability.
Missing related work:
* TyDiQA: https://arxiv.org/pdf/2003.05002
- this is one of the earlier work that collected multilingual questions for factoid questions with native speakers.
* GooAQ: https://arxiv.org/abs/2104.08727
- this paper also uses google to collect QA dataset.

---

### Official Review · Reviewer_VPXh · 2024-11-04

**Soundness:** 2
**Presentation:** 3
**Contribution:** 3
**Rating:** 6
**Confidence:** 4

**Summary:**

This paper introduces a semi-automatic framework for collecting multilingual QA data and provides a new dataset of questions in 7 languages from 9 regional language communities covering 18 topics. The languages in the new data include higher, medium, low, and extremely low resource languages. The proposed framework involves collecting human authored seed queries and finding related question/answer pairs from a search engine, then expanding this set iteratively. There are manual filtering mechanisms to ensure data quality: domain verification removes questions from untrusted domains, and a human annotation process verifies question quality, region relevance, and answer accuracy. The resulting dataset consists of ~64k verified datapoints. The paper provides some analysis of this data

**Strengths:**

The collection of regionally relevant QA data is very important for expanding the capabilities of QA models. Even high resource languages contain lower resource regional variants and topics which need to be studied independently to improve user experiences in those regions; lower resource languages will benefit from the additional data provided here. The human validation steps taken in this work add non-trivial value to the automatically collected data.

**Weaknesses:**

The evaluation metrics used in this work are poor measures of QA system performance. The authors should consider modern metrics such as LLM-as-judge (https://huggingface.co/papers/2306.05685) or PEDANTS (https://arxiv.org/pdf/2402.11161) to give a better sense of existing model performance on the new data.

**Questions:**

none

---

### Meta-Review · Area_Chair_24RX · 2024-12-19

**Metareview:**

All the reviewers pointed to the quality of the paper in terms of the presentation and point to usefulness of the dataset in general sense of having an extra multilingual benchmark. However, I agree with some of the reviewers that it is not clear what this new dateset is evaluating that give us more insight on weakness of LLMs. Specially since some of the models score 80%+ F1 score across all languages on the provided benchmark already.

**Additional Comments On Reviewer Discussion:**

None

---

### Decision · Program_Chairs · 2025-01-22

Reject